# Significance of 5-*S*-Cysteinyldopa as a Marker for Melanoma

**DOI:** 10.3390/ijms21020432

**Published:** 2020-01-09

**Authors:** Kazumasa Wakamatsu, Satoshi Fukushima, Akane Minagawa, Toshikazu Omodaka, Tokimasa Hida, Naohito Hatta, Minoru Takata, Hisashi Uhara, Ryuhei Okuyama, Hironobu Ihn

**Affiliations:** 1Department of Chemistry, Fujita Health University School of Medical Sciences, 1-98 Dengakugakubo, Kutsukake-cho, Toyoake, Aichi 470-1192, Japan; 2Department of Dermatology and Plastic Surgery, Faculty of Life Sciences, Kumamoto University, 1-1-1 Honjo, Chuo-ku, Kumamoto 860-8556, Japan; satoshi.fukushima.tb@gmail.com (S.F.); ihn-der@kumamoto-u.ac.jp (H.I.); 3Department of Dermatology, Shinshu University School of Medicine, 3-1-1 Asahi, Matsumoto, Nagano 390-8621, Japan; akn@shinshu-u.ac.jp (A.M.); d05dsm019@shinshu-u.ac.jp (T.O.); rokuyama@shinshu-u.ac.jp (R.O.); 4Department of Dermatology, Sapporo Medical University School of Medicine, South 1, West 16, Chuo-ku, Sapporo 060-8543, Japan; hidat@sapmed.ac.jp (T.H.); uharah@sapmed.ac.jp (H.U.); 5Department of Dermatology, Toyama Prefectural Central Hospital, 2-2-78 Nishinagae, Toyama, Toyama 930-8550, Japan; hattanao@tch.pref.toyama.jp; 6Department of Dermatology, Okayama University Graduate School of Medicine, Dentistry and Pharmaceutical Sciences, 2-5-1 Shikada-cho, Kita-Ku, Okayama 700-8558, Japan; mtakata@onyx.ocn.ne.jp

**Keywords:** 5-*S*-cysteinyldopa (5SCD), melanoma, anti-programmed cell death protein (PD)-1 antibody, early detection, biomarker, lactate dehydrogenase (LDH), Melanoma inhibitory activity (MIA), S100 calcium-binding protein B (S100B), hepatocyte growth factor (HGF)

## Abstract

Melanoma is one of the most lethal and malignant cancers and its incidence is increasing worldwide, and Japan is not an exception. Although there are numerous therapeutic options for melanoma, the prognosis is still poor once it has metastasized. The main concern after removal of a primary melanoma is whether it has metastasized, and early detection of metastatic melanoma would be effective in improving the prognosis of patients. Thus, it is very important to identify reliable methods to detect metastases as early as possible. Although many prognostic biomarkers (mainly for metastases) of melanoma have been reported, there are very few effective for an early diagnosis. Serum and urinary biomarkers for melanoma diagnosis have especially received great interest because of the relative ease of sample collection and handling. Several serum and urinary biomarkers appear to have significant potential both as prognostic indicators and as targets for future therapeutic methods, but still there are no efficient serum and urinary biomarkers for early detection, accurate diagnosis and prognosis, efficient monitoring of the disease and reliable prediction of survival and recurrence. Levels of 5-*S*-cysteinyldopa (5SCD) in the serum or urine as biomarkers of melanoma have been found to be significantly elevated earlier and to reflect melanoma progression better than physical examinations, laboratory tests and imaging techniques, such as scintigraphy and echography. With recent developments in the treatment of melanoma, studies reporting combinations of 5SCD levels and new applications for the treatment of melanoma are gradually increasing. This review summarizes the usefulness of 5SCD, the most widely used and well-known melanoma marker in the serum and urine, compares 5SCD and other useful markers, and finally its application to other fields.

## 1. Introduction

Melanoma is a malignant tumor that originates from melanocytes. Although melanoma mainly occurs in the skin (cutaneous melanoma), it can also occur in the eyes (uveal melanoma), gastrointestinal tract, oral mucosa and genital tract (mucosal melanoma) [1,2,3,4,5]. The incidence of melanoma has risen at an increasing global rate per year at 2–7% annually [6]. The risk of developing cutaneous melanoma is known to be higher in Caucasian than in Asian or African subjects, which indicates that the development of cutaneous melanoma is closely related to skin pigmentation since melanin, especially eumelanin, has been shown to have a protective function against UV-induced mutagenesis in the skin [7,8,9]. Several important risk factors that have been linked to the development of melanoma have been identified, including environmental factors, such as exposure to ultraviolet (UV) radiation, and host factors, such as family history [10,11]. As melanoma is one of the most serious and malignant cancers worldwide, efficient biomarkers are needed for early detection, efficient monitoring of the disease, and reliable prediction of survival and recurrence. A number of potentially useful melanoma biomarkers have been reported [12,13,14,15,16,17,18,19,20,21,22,23,24,25,26,27,28,29,30,31,32,33,34,35,36,37,38,39,40,41,42,43,44,45,46,47] (Appendix A). Among those, 5-*S*-cysteinyldopa (5SCD) is one of the most widely used and well-known melanoma biomarkers, especially in Japan. This review gives a historical overview of 5SCD as a biomarker of melanoma and describes the current status of 5SCD to monitor disease progression and to predict prognosis and therapy response of melanoma in comparison with other biomarkers. 

## 2. Intermediate Metabolites of Melanin Synthesis as Melanoma Biomarkers

### 2.1. Urinary Excretion and Serum Concentrations of Intermediate Metabolites in Normal Subjects

Two types of melanin, black to brown eumelanin and reddish to yellow pheomelanin, are produced not only in normal melanocytes but also in melanoma cells [48] (Appendix A). Tyrosinase, a specific enzyme produced in melanocytes, catalyzes the oxidation of the amino acid, tyrosine, to dopaquinone, which leads spontaneously to the formation of 5,6-dihydroxyindole (DHI) and its carboxylic acid derivative, 5,6-dihydroxyindole-2-carboxylic acid (DHICA) in the absence of cysteine. These dihydroxyindoles are further oxidized to produce eumelanin [49]. Conversely, the rapid reaction of dopaquinone with cysteine (or glutathione) leads to the production of 5SCD (or 5-*S*-glutathionyldopa) along with other minor isomers [50]. 5SCD is oxidized further by dopaquinone to afford pheomelanin pigments. The major portion of melanin precursors (dihydroxyindoles and cysteinyldopas) may be oxidized to give rise to melanin pigments. On the other hand, minor amounts of those precursors may leak into the bloodstream because of defective membranes surrounding abnormal and incomplete melanosomes in melanoma cells that could facilitate the leakage of 5SCD into the cytosol as partly *O*-methylated derivatives formed by catechol-*O*-methyl transferase (COMT) in the liver, and excreted into the blood or urine [51] (Figure 1). It is thus possible to estimate the progression of melanoma tumor growth by measuring the concentrations of these melanin-related metabolites in the blood or urine [52,53,54]. The urinary excretion of 5SCD was first used as a biochemical marker of metastatic melanoma [55,56]. It was then shown that the level of 5SCD in the plasma correlates with tumor size in B16 melanoma-bearing mice and is also useful to detect metastasis at an early stage of human melanoma cases [56,57,58,59]. On the other hand, the levels of indolic metabolites, such as 5-hydroxy-6-methoxyindole-2-carboxylic acid (5H6MI2C) and 6-hydroxy-5-methoxyindole-2-carboxylic acid (6H5MI2C), may also be significantly elevated [52,58,59]. DHICA, 5H6MI2C, 6H5MI2C and 5SCD are also known to be excreted into the urine by the sulfate or ester glucuronide conjugates [60].

Wakamatsu et al. showed the urinary excretion and serum concentration levels of 5SCD and 6H5MI2C in 33 healthy Japanese subjects (average age 39 years) [61]. The urinary excretion levels of 5SCD and 6H5MI2C were 0.45 ± 0.29 µmol/day and 0.39 ± 0.32 µmol/day, respectively, and their serum concentrations were 4.3 ± 1.8 nmol/L and 3.6 ± 1.8 nmol/L, respectively. Those two markers showed much greater variation in the urine than in the serum. Since levels of those markers in the urine and serum did not exceed 1.5 µmol/day and 10 nmol/L, respectively, the upper limits of normal values of serum and urine levels of 5SCD and 6H5MI2C in healthy Japanese subjects without melanoma was adopted as 10 nmol/L and 1.5 µmol/day, respectively [61]. 

The urinary excretion and serum concentration levels of 5SCD and 6H5MI2C were compared in relation to age. The excretion levels of both 5SCD and 6H5MI2C were significantly decreased in elderly subjects (53–84 years old) compared with middle-aged subjects (33–44 years old) and young subjects (18–23 years old) [61]. This was thought to be due to a decrease in renal clearance with aging. On the other hand, there were no statistically significant differences in the serum concentration levels of either marker among the three age groups.

Since melanin production is affected by the ambient UV radiation, Wakamatsu et al. examined seasonable variations of serum levels of 5SCD and 6H5MI2C in 10 healthy Japanese subjects by measuring those markers every month over a period of 2 years [62]. Levels of 5SCD were higher in early summer and were lower in early winter. The difference in the average levels was approximately two-fold, but among the 240 samples, no individual values exceeded the upper limit of the normal value, 10 nmol/L. The degree of variation in Japanese subjects was not so pronounced as in subjects living in Sweden [52]. Although fair-skinned whites are more vulnerable to sun exposure than Japanese, the serum (plasma) 5*S*CD levels among whites would be clinically useful throughout the year if samples were excluded from the assay when patients had skin erythema after sun exposure. A significant correlation (*p* < 0.02) was observed between 5SCD levels and solar radiation. Levels of 6H5MI2C showed a smaller variation than levels of 5SCD. No correlation was observed between levels of 6H5MI2C and solar radiation.

There is experimental evidence that albino mice excrete 5SCD at the same level as black mice. This suggests that the production of 5SCD in normal mice can occur in cells that don’t have active tyrosinase. It has also been reported that DOPA and 5SCD are present in the hydrolysis products of the liver and kidney, which do not produce melanin pigments [63]. These experimental results suggest that normal levels of 5SCD may be derived largely from protein-bound 5SCD produced by the non-enzymatic oxidation of tyrosine residues in proteins. It has been reported that this final oxidation reaction, by which dopaquinone is produced and reacts with cysteine to produce cysteinyldopa, is non-enzymatically promoted by various biological oxidations such as hydroxy radicals [64]. This non-enzymatic oxidation pathway is also suggested by experimental results that plasma 5SCD concentrations both in tyrosinase-positive and in tyrosinase-negative albino patients are approximately the same as normal values [65]. Furthermore, urinary 5SCD excretion is known not to depend on skin color [66], and there is no correlation between the excretion of 5SCD and skin types or the number of melanocytes. That is, during UVB irradiation, the amount of 5SCD excreted in subjects with skin type II increased prominently compared with subjects with skin types III to IV [66]. This result suggests that the increase in excretion of 5SCD during UV irradiation is due to UV damage to melanocytes rather than the promotion of melanogenesis.

### 2.2. Melanin Intermediate Metabolites in Melanoma Patients

Wakamatsu et al. [67] compared the relationship between pheomelanin and 5SCD in the serum of melanoma patients. In that study, the mean ± SD serum levels of 5SCD in control subjects (*n* = 36), in melanoma patients without recurrence (*n* = 92) and in melanoma patients with metastases (*n* = 24) were 2.7 ± 1.2 nmol/L (median 2.3 nmol/L), 4.0 ± 1.6 nmol/L (median 3.8 nmol/L) and 72 ± 105 nmol/L (median 35 nmol/L), respectively. The serum levels of 4-amino-3-hydroxyphenylalanine (4-AHP, a degradative marker of pheomelanin) in those three groups were 45 ± 21 nmol/L (median 31 nmol/L), 80 ± 75 nmol/L (median 53 nmol/L) and 306 ± 627 nmol/L (median 133 nmol/L), respectively. The serum levels of 4-AHP in patients with metastases (100 samples from 15 patients with progressive disease) correlated well (*r* = 0.887) with serum levels of 5SCD. The 15 patients reported here were either at stage IV at the initial visit (*n* = 5) or progressed from stage II (*n* = 2) or stage III (*n* = 8) to stage IV. All 15 patients eventually died of melanoma. However, the serum pheomelanin level appeared to be less sensitive than 5SCD in detecting distant metastases. The determination of serum levels of 5SCD is considered to have two complications: they fluctuate with UV exposure and patients with amelanotic metastases usually have normal levels of 5SCD [68]. Wakamatsu et al. did not exclude serum samples taken during the summer, and did not pay any particular attention to avoiding UV exposure of the melanoma patients. Although serum levels of 5SCD in Japanese subjects are elevated two-fold during the early summer, this rise is well below the upper limit of the normal reference range [62]. Among the 240 serum samples obtained from 10 healthy subjects every month over the 2 years, none of the 5SCD values exceeded 10 nmol/L. Thus, the fluctuation of 5SCD levels following UV exposure is not a serious problem in Japan. It should be mentioned, however, that the degree of variation in 5SCD levels in Japanese subjects is not so pronounced as in subjects living in Northern Europe [69]. 

Earlier studies have shown that the urinary *O*-methyl derivatives of indole precursors of eumelanin, such as 5-hydroxy-6-methoxyindole, 6-hydroxy-5-methoxyindole and methoxy-derivatives of the corresponding DHICA, are present at higher levels in melanoma patients compared to healthy individuals, and 5-methoxyindole-2-carboxylic acid is an approved biochemical marker for melanoma [61,70,71]. Yamada et al. reported that the pheomelanin and eumelanin precursor metabolites 5SCD and DHICA plus 6H5MI2C are markers for melanoma [72]. Their results confirmed the presence of DHICA in the urine of patients with diagnosed melanoma where urine of melanoma patients with positive metastasis revealed significant amounts of 5SCD and indoles (DHICA plus 6H5MI2C) above 1 µmol/day and 2 µmol/day, respectively. In the case of patients with metastasis-free melanoma, they excreted DHICA into their urine but always at concentrations <1 µmol/day.

Valko-Rokytovská et al. suggested that the following urinary metabolites can significantly contribute to the detection of melanoma: DHICA, vanilmandelic acid, homovanilic acid, tryptophan, 5-hydroxyindole-3-acetic acid and indoxyl sulphate [73]. However, the indole derivatives 6H5MI2C and 5H6MI2C are considered to be less effective than 5SCD because they are very unstable, and the analytical method needed to detect them is complicated.

Serum levels of 5SCD tend to rise relatively early compared to urinary 5SCD. Serum 5SCD is easier to collect and store than urine samples (24-h urine) and can monitor outpatients regularly. Diagnosis is often uncertain when visceral metastases are suspected by examinations, such as chest radiographs, scintigraphy, echography and CT. In such a case, the diagnostic accuracy of metastasis can be improved if the serum level of 5SCD shows an abnormal value. Therefore, it can be said that 5SCD in the serum is the marker that most accurately reflects the melanoma disease state and the degree of progression. However, in melanomas that do not produce melanin, there are cases in which serum and urinary levels of 5SCD show abnormally high levels for the first time only after visceral metastasis, so the development of a clinical method to enhance the melanogenesis before visceral metastasis is awaited in the future.

Why is 5SCD, a precursor of pheomelanin, produced in high concentrations in melanomas producing eumelanin, such as B16 mouse melanoma? One possibility is that tyrosinase that has leaked from melanosomes catalyzes the production of dopaquinone in the cytoplasm and subsequently produces cysteinyldopa. Indeed, the soluble fraction obtained from B16 and Harding-Passey melanomas contained more 5SCD than the fraction in melanosomes [74]. A second possibility is that the melanosomes often have incomplete membranes, making dopaquinone likely to leak into the cytoplasm [51]. Therefore, 5SCD is less valuable as an indicator of melanogenesis in normal melanocytes, but 5SCD is considered to be a good indicator for melanogenesis in melanoma cells and thus for the degree of melanoma progression. The eumelanin precursors DHI and DHICA are considered to be of low value as biochemical markers for melanoma since they are labile to oxidation, and most of the indole is *O*-methylated by COMT [53]. Therefore, the *O*-methyl forms of these indoles may be markers of eumelanin formation instead of 5SCD. In contrast to 5SCD, these *O*-methyl derivatives are known to correlate with the skin types and hair colors of the subjects. Compared to Caucasians, subjects of African background excrete significantly higher amounts of *O*-methyl 6H5MI2C (or 5H6MI2C) [25,75]. It is also known that the urinary excretion of 5SCD and 6H5MI2C increases several-fold during PUVA therapy and they change in parallel [76]. Therefore, 6H5MI2C and 5H6MI2C appear to be good markers for melanogenesis in normal melanocytes [75,77]. 

## 3. SCD as a Marker of Melanoma Progression

Serum levels of 5SCD were measured in 2648 samples taken from 218 Japanese melanoma patients in order to evaluate the usefulness of 5SCD as a progression marker of melanoma [46] (Appendix A). Levels of 5SCD were significantly elevated above the upper limit of the normal range (10 nmol/L) in stage IV melanoma patients, which suggested that elevated 5SCD values might be regarded as a sign of metastatic melanoma. This study included the largest number of patients and samples used to evaluate the clinical significance of serum levels of 5SCD, except for a report by Bánfalvi et al. in which 252 patients were examined [78]. The sensitivity of elevated serum 5SCD values (>10 nmol/L) to detect distant metastasis (stage IV) was 73% in 62 patients with metastatic melanoma, and the specificity in identifying the absence of distant metastasis was 98% in 156 patients with no apparent distant metastases. The sensitivity was improved to 77% when cases of amelanotic melanoma were excluded. Serum levels of 5SCD before surgical excision of tumors were determined for 141 patients [46]. The mean 5SCD value for those 141 patients was 15.7 nmol/L. The median 5SCD values for stage IV melanoma patients was significantly higher than healthy subjects and patients at melanoma stages I–III (*p* < 0.001). Serum levels of 5SCD were found to increase in a stage-dependent manner, especially for stage IV patients who had particularly high values. Stage IV patients with normal values tended to have amelanotic tumors, skin metastases only or hardly detectable metastases. According to an extensive study carried out in Sweden, the sensitivity of urinary levels of 5SCD to detect metastatic melanoma (stage IIIB and IV) was 60% in 161 patients, and the specificity was 98% in 410 patients. The sensitivity of serum levels of 5SCD to detect distant metastasis was reported to be 70% by Bánfalvi et al. [78]. These studies have shown that 5SCD levels often do not increase in patients with amelanotic melanoma [46,79]. On the other hand, it was known that serum 5SCD in mucosal and uveal melanoma show higher levels [79].

Follow-up of melanoma patients until the end stage revealed an exponential increase in the serum levels of 5SCD in most of those patients. The median value of 20.1 nmol/L at the time of visceral metastasis rose to 249 nmol/L shortly before death. Thus, 5SCD values appear to reflect the tumor burden right up to the end stage. The sharp rise in 5SCD levels during the progression of the disease appears to make 5SCD a particularly sensitive marker for monitoring the response to therapy. In summary, in 42 of the 49 melanoma patients (86%) with visceral metastases (stage IVB), an elevation of serum levels of 5SCD above 10 nmol/L was observed. In 19 of the 49 patients, serum 5SCD values were obtained at the onset of skin/lymph node metastasis (stage IIB to IVB) and 12 of those 19 patients (63%) had elevated 5SCD levels. The elevation of serum levels of 5SCD preceded clinical detection of visceral metastases in 16 of the 49 patients with progressive disease (33%), while the elevation coincided with the detection of visceral metastasis in 18 patients (37%). Thus, serum 5SCD values are as effective as conventional physical examinations and imaging techniques in detecting visceral metastases [46]. 

Cumulative survival curves for melanoma patients who underwent surgical resection according to their 5SCD levels pre- and post-operatively were examined [46]. In patients with elevated 5SCD values pre- and post-operatively (*n* = 5), the 50% survival time was 10 months and the cumulative survival was 20%. In patients with 5SCD values above 10 nmol/L pre-operatively and below 10 nmol/L post-operatively (*n* = 15), the 50% survival time was 18 months and the cumulative survival was 23%. In patients with 5SCD values below 10 nmol/L pre- and post-operatively (*n* = 108), the 5 years survival was 76%. Serum levels of 5SCD are also useful as a prognostic marker; melanoma patients with elevated 5SCD values before or after excision of their tumors had significantly shorter survival times compared with those who had normal values. It is interesting to note that pre-operative values of 5SCD exceeding 10 nmol/L were associated with a poor prognosis, even when post-operative values were below the upper limit. Patients without metastases rarely had a 5SCD value exceeding 30 nmol/L. After the 5SCD value rose above 30 nmol/L, patients with distant metastases survived an average of 6.3 months. These results show that 5SCD levels reflect tumor burden sensitively, and that higher 5SCD values might indicate more widespread dissemination of melanoma metastases.

Kärnell et al. reported urinary 5SCD and 6H5MI2C in 92 patients with melanoma during chemotherapy [80]. The sensitivity of 5SCD for the detection of stage III–IV melanoma was 83%, while the corresponding sensitivity of 6H5MI2C was 52%. A significant correlation was found between 5SCD decrease and clinical regression (*p* < 0.001) and between 5SCD increase and clinical progression (*p* < 0.001). Corresponding correlations were not found for 6H5MI2C. Thus, the use of 5SCD was recommended as a valuable, reliable and simple biochemical markers to use in the clinical follow-up of melanoma patients with advanced disease. Johansson et al. showed the use of reverse transcription polymerase chain reaction (RT-PCR) analysis of melanoma specific transcripts for the identification of circulating melanoma cells [81]. Pigment-related and S100 calcium-binding protein B (S100B) transcripts were quantified in 12 different melanoma cell lines and related to the 5SCD levels, pigment and S100B. Tyrosinase, tyrosinase-related protein-1, and tyrosinase-related protein-2 mRNA correlated significantly with 5SCD levels. The amount of S100B mRNA correlated significantly with the amount of S100B (*p* < 0.001). The measurement of S100B mRNA was more sensitive, but the use of this transcript was hampered by its presence in the blood cells.

## 4. Comparison of 5SCD with Other Melanoma Markers

Several of the melanoma markers have been directly compared with 5SCD. Umemura et al. reported that serum levels of 5SCD have a higher sensitivity than serum lactate dehydrogenase (LDH) in advanced stage melanoma, and that in patients with stage III/IV melanoma, a serum level of 5SCD >15.0 nmol/L at their initial hospital visit correlated with a poor prognosis [82]. Simultaneous measurements of serum levels of 5SCD and intercellular adhesion molecule-I (ICAM-1) showed that the level of 5SCD in the serum is a better marker for disease progression when patients with amelanotic melanoma are excluded [83,84]. 

S100B is also a widely used serum marker for melanoma. Two reports have directly compared the efficacy of serum (or urinary) levels of 5SCD with S100B. Kärnell et al. [47] reported that the serum level of S100B has a better association with the survival rate than urinary 5SCD, while Bánfalvi et al. [85] showed that serum S100B seems to be slightly more sensitive than serum 5SCD. Thus, further studies comparing levels of 5SCD with S100B in the serum are needed to explore the significance of these two markers for the early detection of distant metastases. Melanoma inhibitory activity (MIA) is known to be excreted from chondrocytes and from melanoma cells. MIA has been reported to have effects on cell growth and adhesion, and it may play a role in melanoma metastasis and cartilage development [86]. A significant correlation between survival and serum S100B and MIA levels has been reported [87,88]. In patients with stage IV melanoma who responded to chemotherapy, a decreased MIA serum level was observed. MIA levels showed a good correlation with LDH levels in stage IV melanoma (as was the case for S100B), suggesting that serum MIA levels might be influenced by tumor burden. Several papers have suggested that MIA may be sensitive enough to be used for the detection of both clinically apparent and non-apparent metastatic melanoma disease and for monitoring therapy [89,90,91]. Matsushita et al. compared the usefulness of 5SCD and MIA to monitor post-surgical melanoma patients [91]. They showed the clinical course and serum levels of 5SCD and MIA in two representative cases. In case A, an abnormal serum MIA level was noted shortly after surgical resection of the primary tumor that persisted for 46 months, and finally in-transit and lung metastases became clinically apparent. While MIA levels did not represent melanoma tumor burden as accurately as 5SCD, the sustained elevation of MIA levels may indicate the presence of occult residual disease. In case B, elevation of serum MIA was repeatedly recorded from 8 months before regional lymph node metastasis was apparent. After lymph node dissection, abnormal MIA levels persisted until the lung metastases were detected.

Uslu et al. described that S100B and MIA are highly sensitive tumor markers for monitoring of patients with melanoma with current metastases, but less sensitive for monitoring of tumor-free patients. In this study, MIA had a slightly superior sensitivity to detect progressive disease compared to S100B and seems to be more useful in monitoring of patients with metastatic melanoma receiving immunotherapy [92]. Feuerer et al. recently reported that MIA expression is correlated with a senescent state in melanocytes, and induction of replicative or oncogene-induced senescence resulted in increased MIA expression in vitro. This shows for the first time that MIA is a regulator of cellular senescence in human and murine melanocytes [93]. 

Leptin has been identified as a central mediator that regulates energy intake and expenditure, including appetite, metabolism and fat storage [94]. Several studies have reported the relationship between leptin and melanoma. Leptin receptors are known to be expressed by human melanoma cells [95]. Mizutani et al. reported that there was a significant increase in serum leptin receptor levels in patients with melanoma [96]. The serum levels of leptin receptor decreased gradually with the stage of melanoma, being highest at in situ and lowest at stage IV. The rates of decreased serum leptin receptor concentrations under the cut-off value were higher than those of increased serum 5SCD levels in patients with in situ, stage III and stage IV melanomas and in all melanoma patients. These results suggested that serum levels of leptin receptor might serve as a useful biomarker for the detection of melanoma.

Hida et al. described the efficacy of circulating melanoma cells (CMC) to detect metastasis as 5SCD does not respond until the tumor burden becomes high [97]. The sensitivity of CMC and 5SCD for the detection of metastasis was 33 and 50%, respectively. The combination of CMC and 5SCD showed a sensitivity of 67%, the best performance among CMC, 5SCD, LDH and any combination of two of the markers. These data show that CMC may complement the efficacy of 5SCD to detect metastasis and can be a useful prognostic marker. 

## 5. Usefulness of 5SCD in Monitoring Treatments for Melanoma

Along with the expansion of therapeutic options for metastatic melanoma, the development of useful biomarkers is urgently required to predict and monitor treatment response.

Wimmer et al. previously reported that melanoma patients at an advanced stage who had decreased serum levels of 5SCD after immunochemotherapy had significantly longer survival times [98]. Kärnell et al. also found increased proportions of elevated urinary 5SCD excretion in patients with higher numbers of metastases and a significant correlation between decreases in 5SCD and clinical regression during chemotherapy in advanced disease [47]. Sasaki et al. described the multidisciplinary therapy for metastatic primary melanoma of the esophagus (PMME) [70]. They report the first case of recurrent PMME to be treated with combinations of chemotherapy, immuno-therapy, radiotherapy (RT) and laparoscopic lymphadenectomy. This finding indicates that the combination of cytotoxic and molecular-targeted chemotherapy and RT may be suitable for select patients with metastatic PMME. Levels of 5SCD correlated well with tumor size, and the trend was well matched with treatment.

Nakamura et al. examined whether the treatment effect of dacarbazine was correlated with changes in serum levels of LDH and 5SCD in patients who failed to respond to treatment with immune checkpoint inhibitors [99]. During dacarbazine treatment, LDH and 5SCD was reduced or stable in the case of tumor regression or stable disease, but not tumor progression, which showed that they are useful markers to evaluate the treatment effects for selecting the appropriate melanoma treatment.

Omodaka et al. examined serum levels of 5SCD in 12 metastatic melanoma patients [100] and compared 5SCD and LDH levels measured before and after 3–6 weeks of treatment with nivolumab (Nivo) and their changes with the clinical responses. A decrease of 10 nmol/L or more of serum 5SCD was observed only in partial response patients (2/3 cases, 67%), while an increase of 10 nmol/L or more of serum 5SCD was seen only in progressive disease patients (4/8 cases, 50%). Serum 5SCD changes were within ±10 nmol/L in the remaining six patients (partial response, one; stable disease, one; progressive disease, four). The results of the four patients with progressive disease in whom 5SCD changes were within 10 nmol/L were suspected to have been influenced by small-sized metastatic lesions, a mixed response that included diminished and enlarged metastatic lesions, prior therapy to Nivo with B-Raf proto-oncogene serine/threonine kinase (BRAF) inhibitors or radiation, or the development of brain metastasis. Serum levels of 5SCD in the early phase of Nivo treatment may be helpful to predict therapeutic response in metastatic melanoma (Figure 2). In actual clinical practice, it seems better to use a combination of LDH and 5SCD.

In order to investigate whether the serum concentration of hepatocyte growth factor (HGF) can be used as a biomarker of anti-programmed cell death protein (PD)-1 antibodies, Kubo et al. compared serum HGF concentrations in responder and non-responders to anti-PD-1 antibody treatment [101]. HGF was suggested to serve as a paracrine mediator that controls placental development and growth [102]. In addition, HGF also controls the growth, invasion and metastasis of cancer cells and activating mutations of the receptor for HGF (c-Met) predisposes humans to cancer [103]. Figure 3 shows a comparison of serum HGF concentrations between 5SCD and LDH in responders and non-responders to anti-PD-1 antibody. The non-responders had elevated serum concentrations of HGF compared with the responders (*p* = 0.00124). On the other hand, the serum concentrations of 5SCD and LDH also followed this tendency [101]. 5SCD may be superior to LDH in predicting the treatment response and disease progression in patients receiving immune checkpoint inhibitors [82]. Umemura et al. also reported recent developments in novel molecular targeted therapies, such as immune-checkpoint blockades, serine/threonine-protein kinase B-Raf, and mitogen-activated protein kinase inhibition. The prognosis of advanced melanoma has been improved. The study by Umemura et al. compared the utility of serum 5SCD and LDH levels as markers for predicting the effects of Nivo in treating advanced melanoma, and showed that serum 5SCD levels have the potential to be an excellent predictive marker for the efficacy of Nivo therapy in patients with advanced melanoma [104]. In detail, analyzing 12 metastatic melanoma patients under Nivo therapy, they showed that serum 5SCD levels decreased in 2 of 3 patients having a partial response and increased 4 of 8 progressive cases. Their results suggest that possible monitoring efficacy of 5SCD in the early phase of Nivo therapy. The present study showed the utility of 5SCD as predictive marker for the efficacy of long-term Nivo therapy. On the other hand, serum LDH levels > the upper limit of the normal range did not show worse prognosis. Thus, baseline serum 5SCD levels may be potentially superior to LDH in predicting the efficacy of Nivo therapy. Their analysis showed that patients with baseline serum 5SCD levels > 25.0 nmol/L had significantly poor prognosis. In contrast, serum LDH levels at the upper limit of the normal range did not exhibit such changes.

## 6. SCD Levels in Patients with Non-Cutaneous Melanomas

Approximately 30% of patients with cutaneous melanoma develop metastatic disease, and approximately 40% of metastatic lesions are initially found in the lungs [105]. Melanoma usually arises in the skin, but it can sometimes arise in mucosal sites, such as the oral cavity, esophagus, larynx, vagina and anorectal region [106]. Visceral melanoma is relatively unusual, and melanoma originating in the lungs is extremely rare. Shikuma et al. reported a patient with a primary melanoma of the lung using 5SCD levels to monitor the effects of surgery and chemotherapy [107]. Comparing serum 5SCD levels before and after surgery, the serum 5SCD level normalized from 24.9 to 6.3 nmol/L 2 weeks after surgery. Three months after the surgery, the patient had a recurrence. Mediastinal nodules and a pericardial effusion were seen in a computed tomographic scan and the serum level of 5SCD increased to 345 nmol/L. The patient was given chemotherapy consisting of three cycles of a dacarbazine, nimustine hydrochloride and vincristine regimen (also known as a DAV regimen). A computed tomographic scan performed after three cycles of chemotherapy confirmed a partial response; there was improvement in the pericardial effusion and in the mediastinal nodules. The patient’s serum 5SCD level had decreased to 36.7 nmol/L after 1 cycle of chemotherapy. After 2 cycles of chemotherapy, the serum 5SCD level had decreased to 12.6 nmol/L. The authors concluded that serum 5SCD is a very useful marker for monitoring the progression of primary melanoma of the lung, an extremely rare disease.

Aoki et al. described a case report of melanoma in the male urethra with an increased serum 5SCD concentration [108]. Melanoma of the male urethra is uncommon and the prognosis is favorable if it is detected in an early stage. This case report suggested that the measurement of serum 5SCD is useful for an early diagnosis of male urethral melanoma.

Goto et al. examined 5SCD levels in the serum of 16 patients with primary uveal melanomas [109]. Serum levels of 5SCD in patients with extraocular metastases of melanoma were significantly elevated. Levels of 5SCD in intraocular fluids, especially the vitreous humor, were higher in patients with uveal melanoma than in normal subjects.

## 7. Other Applications Influencing the 5SCD Levels

Diffuse pigmentation is common among patients with chronic renal failure who are undergoing hemodialysis (HD) [110]. The mechanism(s) of skin hyperpigmentation in HD patients remain unclarified, but the accumulation of middle-molecular-weight molecules, such as lipochromes, urochromic pigments and α-melanocyte-stimulating hormone, has been implicated [111]. In chronic renal failure as well as in other diseases, oxidative stress results from the oxidative interaction of proteins, lipids, carbohydrates, DNA and antioxidant substrates with reactive oxygen species such as hydroxyl radicals, superoxide radicals and hydrogen peroxide [112,113,114]. Regarding the chronic renal failure, Murakami et al. described that serum levels of 5SCD were elevated 9-fold in chronic renal failure undergoing hemodialysis (HD) patients compared with controls. They showed that serum levels of free 5SCD and pheomelanin were significantly elevated in the sera of HD patients compared with healthy controls, suggesting the possibility that pheomelanin accumulates in the skin of HD patients due to the high circulating levels of 5SCD, and that oxidative stress promotes these oxidative process [115]. Protein-bound 3,4-dihydroxyphenylalanine (PB-DOPA) is formed when hydroxyl radicals react with tyrosine residues in proteins [116]. Further oxidation of PB-DOPA by hydroxyl radicals may lead to the production of protein-bound 5SCD (PB-5SCD) by combining DOPAquinone residues with cysteine residues [64]. Some HD patients were found to have elevated levels of PB-DOPA [114]. The relation between skin colors and serum 5SCD levels in control subjects and in HD patients were summarized [117]. Serum 5SCD levels in HD patients were found to significantly correlate with skin color as measured by the melanin index (MI) and the erythema index (EI) values on the upper inner arm and by EI value on the forehead. Pigmentation normally takes place within melanosomes, specific organelles present in melanocytes, and proceeds under the control of many pigmentary factors, including tyrosinase [118]. However, under some pathophysiological conditions, such as UV exposure, pigmentation may also proceed by the action of reactive oxygen species on melanin precursors [119].

Nagano et al. reported a case of melanoma increasing during pregnancy by measuring 5SCD in tissue [120]. 5SCD was measured in the primary resected melanoma tissue and the post-operative scar formed after the primary excision. The primary nodular lesion level was as high as 73.40 ng/mg, and level in the adjacent tissue 2.5 cm away from the edge of the primary tumor was 0.70 ng/mg. They concluded that measuring 5SCD levels in the tissues would be useful for early detection of small metastatic foci or confirmation of residual foci.

As mentioned above, it was known that levels of 5SCD were elevated by UV irradiation, in pregnant women or in renal failure patients, but Yoshino et al. reported that 5SCD were also elevated by the intake of mushroom or agaricus (*Agaricus subrufescens*) [121].

## 8. Discussion and Future Perspectives

As the pathways for the future progress of this 5SCD marker, it is necessary to examine the advanced technology to evaluate the correlation between cut-off values with LDH, S100B, MIA, leptin, and so on, and to evaluate as much circulating markers as possible from serum and urine. In addition, the measurement of samples from other than serum or urine, for example, in the cerebrospinal fluid (CSF) or 5SCD levels of BRAF negative versus BRAF positive patients may be a new development. Shinno et al. measured 5SCD levels in CSF in a patient with a variant of neurocutaneous melanosis with leptomeningeal melanoma and reported the clinical significance of 5SCD [122]. Combined chemoimmunotherapy was repeated every other month with monitoring of 5SCD level in CSF. 5SCD levels in CSF have not been monitored in patients with neurocutaneous melanosis, so the basal 5SCD levels in CSF were unclear. By measuring 5SCD levels in CSF collected from patients from Parkinson disease, hemifacial spasm, and moyamoya disease, the basal 5SCD in CSF was determined as 0 nmol/L. 5SCD level decreased after each treatment, but the basal level steadily increased prior to the next treatment. Their findings suggested that periodic measurement of 5SCD levels in CSF is valuable for evaluating the therapeutic efficacy of chemoimmunotherapy in patients with malignant leptomeningeal melanoma and for monitoring the progression of melanoma. On the other hand, Wakamatsu et al. measured 5SCD level in CSF from one patient. 5SCD levels in serum and CSF were 11.0 nmol/L and 513 nmol/L, respectively (unpublished data). The patient was hospitalized by suspicion of meningoencephalitis in the neurology department. The infection was negative, and melanoma was suspected by a cytology of CSF and consulted by a dermatologist. Therefore, 5SCD determination in serum and CSF was probably thought to be contributed to the diagnosis. There was no primary in skin, and no visceral metastasis, and melanoma was diagnosed as unknown primary region at that time. It was thought to be a melanoma originating in the central nervous system. Circulating cell-free microRNAs (miRNAs) are thought to be involved in the regulation of gene expression and vary with cancer progression or therapeutic response [123]. Using serum samples from 33 patients with malignant melanoma before anti-PD-1 antibody therapy (Nivo or pembrolizumab), Nakahara et al. measured several miRNAs [124]. Their results suggest that miR-16-5p, miR-17-5p, and miR-20a-5p are useful indicators of successful anti-PD-1 antibody therapy. Furthermore, they proposed that evaluating serum miRNA levels may be a simple method to predict the effectiveness of immune checkpoint inhibitors, although the numbers of patients evaluated are small. To determine whether these three miRNAs can be worthy biomarkers for anti-PD-1 therapy, further studies with a higher number of patients are required.

BRAF/MEK inhibitors are generally considered to be equally effective whether given before or after immunotherapy. In BRAF mutated patients with normal LDH, first-line immunotherapy seems to be a more effective approach. Czarnecka et al. has demonstrated that the use of two different systemic treatment modalities allows the achievement of comparable survival in BRAF mutated and BRAF wild-type patients although BRAF mutation is a negative prognostic factor in stage IV melanoma [125].

Koh et al. described about primary malignant melanoma of the breast (PMMB) [126]. PMMB is a rare tumor with only a few case reports available in the literature. They reported two cases of PMMB, one derived from the breast parenchyma and the other from the breast skin. The first case consisted of atypical epithelioid cells without overt melanocytic differentiation like melanin pigments. The tumor cells showed diffuse positivity for S100B, tyrosinase, and BRAF V600E. However, the tumor cells were negative for cytokeratin, epithelial membrane antigen, and human melanoma black-45 (HMB-45). The second case showed atypical melanocytic proliferation with heavy melanin pigmentation. The tumor cells were positive for S100B, HMB-45, tyrosinase, and BRAF V600E. Although there is little evidence for breast in these two cases, these two cases represent two distinct presentations of PMMB in terms of skin involvement, melanin pigmentation, and HMB-45 positivity.

## 9. Conclusions

Melanoma is one of the most lethal cancers worldwide, but still there are not efficient serum biomarkers to conduct an early detection and an efficient monitoring of the disease. Therefore, it is of utmost importance to discover novel circulating markers in order to implement diagnosis, prognosis and treatment of this malignancy. This review confirms 5SCD as an excellent candidate due to its potential informative power about important tumor features, the theoretical possibility to use it as an early-stage melanoma marker and the advantages of the detection method compared to biopsies. The serum level of 5SCD is a sensitive and specific marker for predicting the presence of distant melanoma metastases when analyzed regularly. The elevation of serum levels of 5SCD to >10 nmol/L often precedes the clinical detection of visceral metastases. This method is therefore useful in following up outpatients with suspected metastatic melanoma when combined with physical examinations and the routine use of clinical laboratory tests. Serum levels of 5SCD could be used to accurately assess therapy responses in future clinical trials for the treatment of advanced melanoma.

## Figures and Tables

**Figure 1 ijms-21-00432-f001:**
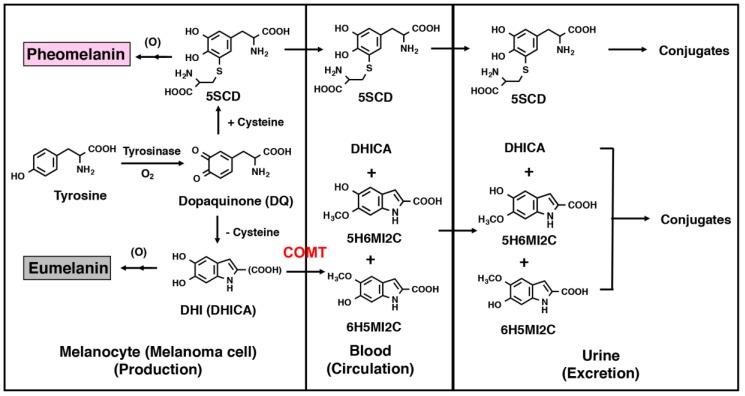
Production, circulation, and excretion of melanin-related metabolites. 5-*S*-cysteinyldopa (5*S*CD); 5,6-dihydroxyindole (DHI); 5,6-dihydroxyindole-2-carboxylic acid (DHICA); catechol-*O*-methyl transferase (COMT); 5-hydroxy-6-methoxyindole-2-carboxylic acid (5H6MI2C); 6-hydroxy-5-methoxyindole-2-carboxylic acid (6H5MI2C).

**Figure 2 ijms-21-00432-f002:**
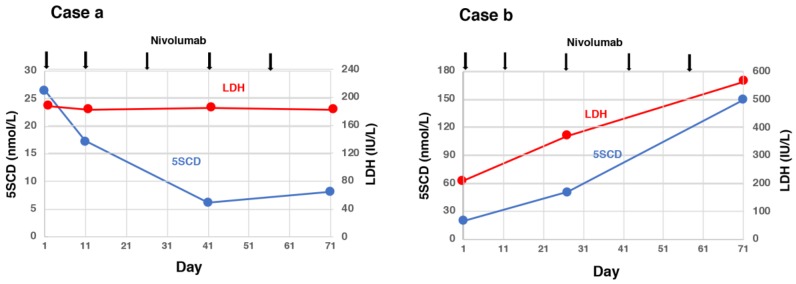
Serum 5SCD and lactate dehydrogenase (LDH) levels at baseline and treatment with nivolumab (Nivo). Case a—Serum 5SCD at 41 days after the first administration of Nivo was decreased, while serum LDH remained stable. The result of Nivo was determined as a partial response. Case b—Serum 5SCD and LDH at 27 days after the first administration of Nivo were highly increased. The result of Nivo was determined as progressive disease. This Figure was modified with the consent of authors (Omodaka et al., [100]).

**Figure 3 ijms-21-00432-f003:**
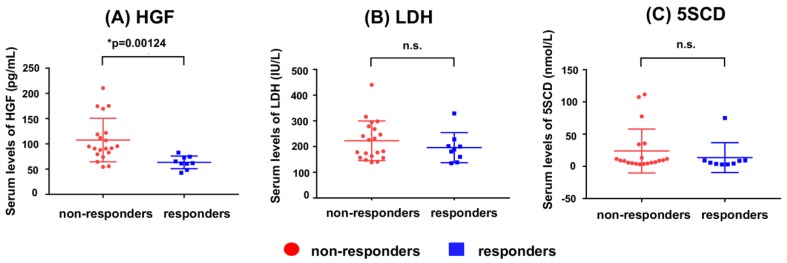
The serum concentration of hepatocyte growth factor (HGF) (**A**), LDH (**B**) and 5SCD (**C**) in non-responders and in responders to anti-PD-1 therapy. This Figure was modified with the consent of authors (Kubo et al. [101]).

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
