# Peer review of "Significance of 5-S-Cysteinyldopa as a Marker for Melanoma"

_ijms, 2020, doi:10.3390/ijms21020432_

Round 1

Reviewer 1 Report

5-S-cysteinyldopa (5SCD) has been used as a marker for melanoma progression in some studies. However, various factors can elevate the serum level of 5SCD (sun exposure, renal failure, chemotherapies and some drugs) and this should be underlined. In case of amelanotic melanoma, as reported, usually there are normal levels of 5SCD. What about mucosal melanoma and uveal melanoma?

This is an interesting and well-written review about the melanoma and 5SCD. The identification of prognostic factors and biomarkers that predict response to treatment is essential in order to improve survival in patients with melanoma. To this aim, a review should analyze all the prognostic factors discussed in the literature. The serum lactate dehydrogenase, the serum level of S100 protein and the melanoma inhibitory activity were reported only In comparison of 5SCD.

Perhaps it may also be useful to discuss other prognostic factors that influence recent studies, such as miRNA and target genes that can represent important prognostic and therapeutic biomarkers.

Author Response

Dear Reviewer 1:

I greatly appreciate your valuable comments. The authors have added some corrections in the revised paper according to your comments.

1. 5-S-cysteinyldopa (5SCD) has been used as a marker for melanoma progression in some studies. However, various factors can elevate the serum level of 5SCD (sun exposure, renal failure, chemotherapies and some drugs) and this should be underlined.

 <Answer>The authors have already described in this paper that it is known that levels of 5SCD are elevated not only by UV irradiation, but also by the intake of mushroom or agaricus (Agaricus subrufescens), the renal failure, and also in pregnant women, but as the reviewer 2 suggested to delete the section ‘7. Other applications of 5SCD in addition to melanoma’ in which the sentence about the renal failure was described. Thus, we added the following sentence.

Regarding the chronic renal failure, Murakami et al. described that serum levels of 5SCD were elevated 9-fold in chronic renal failure undergoing hemodialysis patients compared with controls. They showed that serum levels of free 5SCD and pheomelanin were significantly elevated in the sera of hemodialysis (HD) patients compared with healthy controls, suggesting the possibility that pheomelanin accumulates in the skin of HD patients due to the high circulating levels of 5SCD, and that oxidative stress promotes these oxidative process [106].

2. In case of amelanotic melanoma, as reported, usually there are normal levels of 5SCD. What about mucosal melanoma and uveal melanoma?

<Answer>The author added the following sentence. On the other hand, it was known that serum 5SCD in mucosal and uveal melanoma show higher levels (76).

3. This is an interesting and well-written review about the melanoma and 5SCD. The identification of prognostic factors and biomarkers that predict response to treatment is essential in order to improve survival in patients with melanoma. To this aim, a review should analyze all the prognostic factors discussed in the literature. The serum lactate dehydrogenase, the serum level of S100 protein and the melanoma inhibitory activity were reported only in comparison of 5SCD. Perhaps it may also be useful to discuss other prognostic factors that influence recent studies, such as miRNA and target genes that can represent important prognostic and therapeutic biomarkers.

<Answer>We added two papers regarding miRNA.

Circulating cell-free microRNAs (miRNAs) are thought to be involved in the regulation of gene expression and vary with cancer progression or therapeutic response [107]. Using serum samples from 33 patients with malignant melanoma before anti-programmed cell death protein 1 (anti-PD-1) antibody therapy (nivolumab or pembrolizumab), Nakamura et al. measured several microRNAs (miRNAs)[108]. Their results suggest that miR-16-5p, miR-17-5p, and miR-20a-5p are useful indicators of successful anti-PD-1 antibody therapy. Furthermore, they proposed that evaluating serum miRNA levels may be a simple method to predict the effectiveness of immune checkpoint inhibitors, although the numbers of patients evaluated are small. To determine whether these three miRNAs can be worthy biomarkers for anti-PD-1 therapy.

107. Polini, B.; Carpi, S.; Romanini, A.; Breschi, M.; Nieri, P.; Podesta, A. Circulating cell-free microRNAs in cutaneous melanoma staging and recurrence or survival prognosis, Pigment Cell Melanoma Res. 2019, 32, 486–499.

108. Nakahara, S.; Fukushima, S.; Okada, E.; Morinaga, J.; Kubo, Y.; Tokuzumi, A.; Matsumoto, S.; Tsuruta-Kadohisa, M.; Kimura, T.; Kuriyama, H.; Miyashita, A.; Kajihara, I.; Jinnin, M.; Ihn, H. MicroRNAs that predict the effectiveness of anti-PD-1 therapies in patients with advanced melanoma. J. Dermatol. Sci. 2019. doi:10.1016/j.jdermsci.2019.11.010.

Reviewer 2 Report

The review entitled Significance of 5-S-cysteinyldopa as a marker for melanoma by Wakamatsu et al highlights the importance of this serum/urinary marker in cutaneous melanoma.

The review is well written but it can beneficiate from some additions.

Comments:

As it is a review a drawing/scheme of the synthesis pathway of eumelanin and pheomelanin would be helpful for the reader to establish the “biochemical position” of 5SCD.

Page 3, row 137 – please specify what are the actual 100 samples harvested from 15 patients?

Page 4, row 156-157 – please correct “…the presence of DHICA in the urine of patients with diagnosed melanoma where metastases were present at concentrations >1 μmol/day.”

Page 4, row 172 - Please elaborate on the following phrase that seems strange “…clinical method to enhance melanin production in melanomas that do not produce melanin is awaited in the future.”

Row 184 – please specify what is the full name of COMT

Row 192 – Yet again, please specify what are the 2,648 samples taken from 218 Japanese? Serum/urine samples taken in dynamics, in melanoma clinical evolution, before and after excision, during therapy?

Page 6, row 255 and onward please use more recent references as there are important studies on serum MIA and S100, to mention only the Erlangen University group papers.

Row 261 – please replace “Several recent papers..” as the mentioned references (84-86) are almost 19 years old. See my above comment.

Row 292 – unless the references have an historical value for the issue, please replace ref 90 as it is a 1997 publication where chemotherapy was still used in melanoma.

Row 305 – regarding the work of ref 91, can you please elaborate on how the changes in serum LDH and 5-S-cysteinyldopa could be used to select actually the non-responders from the responders to nivo/ipi treatment?

Row 336 – the same comment as above for the work published in ref 96

Row 364 – my suggestion is to delete this section as it does not match the title of the review and instead give the reader a section of future development pathways for the evaluation of this marker: enlarging the number of patients, correlating its cut-offs with that of LDH, S100, MIA, leptin and so on; using up-dated technology for evaluating as much circulating markers as possible from serum, urine, maybe CFL from brain metastases, BRAF poz versus BRAF neg patients levels of this marker……

General comments

Please correct the editing mistakes, such as the ones in Page 2, Row 51 or Page 3, row 109, row 301 and so on.

From a total of 122 references, 22 are of the group that is submitting this paper. It is OK to cite your work but other groups, for example Karnell group evaluating cysteinyldopa for the follow-up is not cited.

In the acknowledgement section – as it is a review paper what is the reason to acknowledge “Ms. Yukiko Nakanishi for the measurements of 5SCD”.

Author Response

Dear Reviewer 2:

I greatly appreciate your valuable comments. The authors have added some corrections in the revised paper according to your comments.

The review entitled Significance of 5-S-cysteinyldopa as a marker for melanoma by Wakamatsu et al highlights the importance of this serum/urinary marker in cutaneous melanoma.

The review is well written but it can beneficiate from some additions.

Comments:

As it is a review a drawing/scheme of the synthesis pathway of eumelanin and pheomelanin would be helpful for the reader to establish the “biochemical position” of 5SCD.

<Answer> I added the figure in Supplemental Figure S1.

Page 3, row 137 – please specify what are the actual 100 samples harvested from 15 patients?

<Answer> The 15 patients reported here were either at stage IV at the initial visit (n = 5) or progressed from stage II (n =2) or stage III (n =8) to stage IV. All 15 eventually died of melanoma.

Page 4, row 156-157 – please correct “…the presence of DHICA in the urine of patients with diagnosed melanoma where metastases were present at concentrations >1 μmol/day.”

<Answer> We changed the sentence as follows:

Urine of melanoma patients with positive metastasis revealed significant amounts of 5SCD and indoles (DHICA plus 6H5MI2C) above 1 μmol/day and 2 μmol/day, respectively.

Page 4, row 172 - Please elaborate on the following phrase that seems strange “…clinical method to enhance melanin production in melanomas that do not produce melanin is awaited in the future.”

<Answer> Thank you very much for your comment. I changed as follows:

The development of clinical method to enhance the melanogenesis before visceral metastasis is awaited in the future.

Row 184 – please specify what is the full name of COMT

<Answer> COMT is catechol-o-methyl transferase

Row 192 – Yet again, please specify what are the 2,648 samples taken from 218 Japanese? Serum/urine samples taken in dynamics, in melanoma clinical evolution, before and after excision, during therapy?

<Answer> Regarding this comments, I added two tables in Supplemental TableS2, and Table S3.

Page 6, row 255 and onward please use more recent references as there are important studies on serum MIA and S100, to mention only the Erlangen University group paper.

<Answer> S100 and MIA are highly sensitive tumor markers for monitoring of patients with melanoma with current metastases, but less sensitive for monitoring of tumor-free patients. In the current study, MIA had a slightly superior sensitivity to detect progressive disease compared to S100 and seems to be more useful in monitoring of patients with metastatic melanoma receiving immunotherapy [89]. Feuerer et al. recently reported that MIA expression is correlated with a senescent state in melanocytes, and Induction of replicative or oncogene‐induced senescence resulted in increased MIA expression in vitro. This shows for the first time that MIA is a regulator of cellular senescence in human and murine melanocytes [90].

Uslu, U.; Schliep, S.; Schliep, K.; Erdmann, M.; Koch, H. U.; Parsch, H.; Rosenheinrich, S.; Anzengruber, D.; Bosserhoff, A. K.; Schuler, G.; Schuler-Thurner, B. Comparison of the Serum Tumor Markers S100 and Melanoma-inhibitory Activity (MIA) in the Monitoring of Patients with Metastatic Melanoma Receiving Vaccination Immunotherapy with Dendritic Cells. Anticancer Res., 2017, 37, 5033-5037. Feuerer, L.; Lamm, S.; Henz, L.; Kappelmann-Fenzl, M.; Haferkamp, S.; Meierjohann, S.; Hellerbrand, C.; Kuphal, S.; Bosserhoff, A. K. Role of melanoma inhibitory activity in melanocyte senescence. Pigment Cell Mel. Res., 2019, 32, 777-791. doi: 10.1111/pcmr.12801.

Row 261 – please replace “Several recent papers..” as the mentioned references (84-86) are almost 19 years old. See my above comment.

<Answer> I added the proper two references [89, 90].

Row 292 – unless the references have an historical value for the issue, please replace ref 90 as it is a 1997 publication where chemotherapy was still used in melanoma.

<Answer> As this paper has an historical value, I would like to use this paper without deleting it. Instead, I added the word ‘previously’.

Row 305 – regarding the work of ref 91, can you please elaborate on how the changes in serum LDH and 5-S-cysteinyldopa could be used to select actually the non-responders from the responders to nivo/ipi treatment?

<Answer> During dacarbazine treatment, LDH and 5-S-cysteinyldopa was reduced or stable in the case of tumor regression or stable disease, but not tumor progression. We have added the description on lines 349-352 of revised manuscript.

Nakamura et al. examined whether the treatment effect of dacarbazine was correlated with changes in serum levels of LDH and 5SCD in patients who failed to respond to treatment with immune checkpoint inhibitors [95]. During dacarbazine treatment, LDH and 5-S-cysteinyldopa was reduced or stable in the case of tumor regression or stable disease, but not tumor progression, which showed that they are useful markers to evaluate the treatment effects for selecting the appropriate melanoma treatment.

Row 336 – the same comment as above for the work published in ref 96

<Answer> We added the following sentence.

In detail, analyzing 12 metastatic melanoma patients under Nivo therapy, they showed that serum 5SCD levels decreased in 2 of 3 patients having a partial response and increased 4 of 8 progressive cases. Their results suggest that possible monitoring efficacy of 5SCD in the early phase of Nivo therapy. The present study showed the utility of 5SCD as predictive marker for the efficacy of long-term Nivo therapy. On the other hand, serum LDH levels > the upper limit of the normal range did not show worse prognosis. Thus, baseline serum 5SCD levels may be potentially superior to LDH in predicting the efficacy of Nivo therapy. Their analysis showed that patients with baseline serum 5SCD levels > 25.0 nmol/L had significantly poor prognosis. In contrast, serum LDH levels at the upper limit of the normal range did not exhibit such changes.

Row 364 – my suggestion is to delete this section as it does not match the title of the review and instead give the reader a section of future development pathways for the evaluation of this marker: enlarging the number of patients, correlating its cut-offs with that of LDH, S100, MIA, leptin and so on; using up-dated technology for evaluating as much circulating markers as possible from serum, urine, maybe CFL from brain metastases, BRAF poz versus BRAF neg patients levels of this marker……

<Answer> As the pathways for the future progress of this 5SCD marker, it is necessary to examine the advanced technology to evaluate the correlation between cut-off values with LDH, S100, MIA, leptin, and so on, and to evaluate as much circulating markers as possible from serum and urine. In addition, the measurement of samples from other than serum or urine, for example, in the cerebrospinal fluid (CSF) or 5SCD levels of BRAF negative versus BRAF positive patients may be a new development. Shinno et al. measured 5SCD levels in the CSF in a patient with a variant of neurocutaneous melanosis with leptomeningeal melanoma, and reported the clinical significance of 5SCD [109]. Combined chemoimmunotherapy was repeated every other month with monitoring of 5SCE level in CSF. CSF levels of 5SCD have not been monitored in patients with neurocutaneous melanosis (NCM), so the basal 5SCD levels in CSF were unclear. By measuring 5SCD in CSF collected from patients from Parkinson disease, hemifacial spasm, and moyamoya disease, the basal 5SCD in CSF was determined as 0 nmol/L. 5SCD level decreased after each treatment, but the basal level steadily increased prior to the next treatment. Their findings suggested that periodic measurement of 5SCD in CSF is valuable for evaluating the therapeutic efficacy of chemoimmunotherapy in patients with malignant leptomeningeal melanoma and for monitoring the progression of the melanoma. On the other hand, Wakamatsu et al. measured 5SCD level in cerebrospinal fluid from one patient. 5SCD levels in serum and cerebrospinal fluid were 11.0 nmol/L and 513 nmol/L, respectively (unpublished data). The patient was hospitalized by suspicion of meningoencephalitis in the neurology department. The infection was negative, and melanoma was suspected by a cytology of the cerebrospinal fluid and consulted by a dermatologist. Therefore, 5SCD determination in serum and cerebrospinal fluid was probably thought to be contributed to the diagnosis. There was no primary in skin, and no visceral metastasis, and melanoma was diagnosed as unknown primary region at that time. It was thought to be a melanoma originating in the central nervous system. BRAF/MEK inhibitors are generally considered to be equally effective whether given before or after immunotherapy. In BRAF mutated patients with normal LDH, first-line immunotherapy seems a more effective approach. Czarnecka et al. has demonstrated that the use of two different systemic treatment modalities allows achievement of comparable survival in BRAF mutated and BRAF wild-type patients although BRAF mutation is a negative prognostic factor in stage IV melanoma [110]. Koh et al. described about Primary malignant melanoma of the breast (PMMB) [111]. PMMB is a rare tumor with only a few case reports available in the literature. They reported two cases of PMMB, one derived from the breast parenchyma and the other from the breast skin. The first case consisted of atypical epithelioid cells without overt melanocytic differentiation like melanin pigments. The tumor cells showed diffuse positivity for S100 protein, tyrosinase, and BRAF V600E. However, the tumor cells were negative for cytokeratin, epithelial membrane antigen, and human melanoma black-45 (HMB-45). The second case showed atypical melanocytic proliferation with heavy melanin pigmentation. The tumor cells were positive for S100 protein, HMB-45, tyrosinase, and BRAF V600E. Although there is little evidence for breast in these two cases, these two cases represent two distinct presentations of PMMB in terms of skin involvement, melanin pigmentation, and HMB-45 positivity.

Shinno, K.; Nagahiro, S.; Uno, M.; Kannuki, S.; Nakaiso, M.; Sano, N.; Horiguchi, H. Neurocutaneous melanosis associated with malignant leptomeningeal melanoma in an adult: clinical significance of 5-S-cysteinyldopa in the cerebrospinal fluid. Neurol. Med. Chir. (Tokyo), 2003, 43, 619-625. Czarnecka, A. M.; Teterycz, P.; Mariuk-Jarema, A.; Lugowska, I.; Rogala, P.; Dudzisz-Sledz, M.; Swaitaj, T.; Rutkowski, P. Treatment sequencing and clinical outcomes in BRAF-positive and BRAF-negative unresectable and metastatic melanoma patients treated with new systemic therapies in routine practice. Target. Oncol. 2019, 14, 729-742. Koh, J.; Lee, J.; Jung, S. Y.; Kang, H. S.; Yun, T.; Kwon, Y. Primary malignant melanoma of the breast: a report of two cases. J. Pathol. Transl. Med. 2019, 53, 119-124.

General comments

Please correct the editing mistakes, such as the ones in Page 2, Row 51 or Page 3, row 109, row 301 and so on.

<Answer> Many thanks for your suggestion. The authors changed the sentences according to Reviewer’s suggestions.

From a total of 122 references, 22 are of the group that is submitting this paper. It is OK to cite your work but other groups, for example Karnell group evaluating cysteinyldopa for the follow-up is not cited.

<Answer> I added two papers of Kårnell et al. and Johansson et al.

Kärnell et al. reported urinary 5SCD and 6H5MI2C in 92 patients with melanoma during chemotherapy [77]. The sensitivity of 5SCD for the detection of stage III-IV melanoma was 83%, while the corresponding sensitivity of 6H5MI2C was 52%. A significant correlation was found between 5SCD decrease and clinical regression (P < 0.001) and between 5SCD increase and clinical progression (P < 0.001). Corresponding correlations were not found for 6H5MI2C. The use of 5SCD was recommended as a valuable, reliable and simple biochemical markers to use in the clinical follow-up of melanoma patients with advanced disease. Johansson et al. showed the use of reverse transcription polymerase chain reaction (RP-PCR) analysis of melanoma specific transcripts for the identification of circulating melanoma cells [78]. Pigment-related and S100B (S100-β) transcripts were quantified in 12 different melanoma cell lines and related to the 5SCD levels, pigment and S100B protein. Tyrosinase, tyrosinase-related protein-1, and tyrosinase-related protein-2 mRNA correlated significantly with 5SCD levels. The amount of S100B mRNA correlated significantly with the amount of S100B protein (P < 0.001). The measurement of S100B mRNA was more sensitive, but the use of this transcript was hampered by its presence in the blood cells.

Kärnell, R.; Kågedal ,B.; Lindholm, C.; Nilsson, B.; Arstrand, K.; Ringborg, U. The value of cysteinyldopa in the follow-up of disseminated malignant melanoma. Melanoma Res., 2000, 10, 363-369. Johansson, M.; takasaki, A.; Lenner, L.; Arstrand, K.; Kågedal, B. Quantitative relationship between pigment-related mRNA and biochemical melanoma markers in melanoma cell lines. Melanoma Res. 2002, 12, 193-200.

In the acknowledgement section – as it is a review paper what is the reason to acknowledge “Ms. Yukiko Nakanishi for the measurements of 5SCD”.

I deleted the name of Ms. Nakanishi.

Round 2

Reviewer 1 Report

The authors addressed my comments appropriately.

The discussion on pages 9 (from line 399) and 10 should be divided into "elements influencing the 5SCD levels", "discussion and future perspectives" and "conclusion". This could help reading the text.

implement the part on the elements that can modify the levels of 5SCD (renal failure ecc)

Author Response

The discussion on pages 9 (from line 399) and 10 should be divided into "elements influencing the 5SCD levels", "discussion and future perspectives" and "conclusion". This could help reading the text.

implement the part on the elements that can modify the levels of 5SCD (renal failure ecc)

<Answer>

Many thanks for your valuable comments. I changed as follows:

I added newly the section ‘7. Other applications influencing the 5SCD levels’ and added the following sentences in this section.

Diffuse pigmentation is common among patients with chronic renal failure who are undergoing hemodialysis (HD) [106]. The mechanism(s) of skin hyperpigmentation in HD patients remain unclarified, but the accumulation of middle-molecular-weight molecules, such as lipochromes, urochromic pigments and α-melanocyte-stimulating hormone, has been implicated [107]. In chronic renal failure as well as in other diseases, oxidative stress results from the oxidative interaction of proteins, lipids, carbohydrates, DNA and antioxidant substrates with reactive oxygen species such as hydroxyl radicals, superoxide radicals and hydrogen peroxide [108-110]. Regarding the chronic renal failure, Murakami et al. described that serum levels of 5SCD were elevated 9-fold in chronic renal failure undergoing hemodialysis patients compared with controls. They showed that serum levels of free 5SCD and pheomelanin were significantly elevated in the sera of hemodialysis (HD) patients compared with healthy controls, suggesting the possibility that pheomelanin accumulates in the skin of HD patients due to the high circulating levels of 5SCD, and that oxidative stress promotes these oxidative process [111]. Protein-bound 3,4-dihydroxyphenylalanine (PB-DOPA) is formed when hydroxyl radicals react with tyrosine residues in proteins [112]. Further oxidation of PB-DOPA by hydroxyl radicals may lead to the production of protein-bound (PB)-5SCD (PB-5SCD) by combining DOPAquinone residues with cysteine residues [59]. Some HD patients were found to have elevated levels of PB-DOPA [110]. The relation between skin colors and serum 5SCD levels in control subjects and in HD patients were summarized [113]. Serum 5SCD levels in HD patients were found to significantly correlate with skin color as measured by the melanin index (MI) and the erythema index (EI) values on the upper inner arm and by EI value on the forehead. Pigmentation normally takes place within melanosomes, specific organelles present in melanocytes, and proceeds under the control of many pigmentary factors, including tyrosinase [114]. However, under some pathophysiological conditions, such as UV exposure, pigmentation may also proceed by the action of reactive oxygen species on melanin precursors [115].

Nagano et al. reported a case of melanoma increasing during pregnancy by measuring 5SCD in tissue [116]. 5SCD was measured in the primary resected melanoma tissue and the post-operative scar formed after the primary excision. The primary nodular lesion level was as high as 73.40 ng/mg, and level in the adjacent tissue 2.5 cm away from the edge of the primary tumor was 0.70 ng/mg. Interestingly, the post-operative scar formed 14 days after the primary excision showed a level of 4.75 ng/mg, which was clearly higher than those of the control normal skin tissues. They concluded that measuring 5SCD levels in tissues would be useful for early detection of small metastatic foci or confirmation of residual foci.

As mentioned above, it was known that levels of 5SCD were elevated by UV irradiation, in pregnant women or in renal failure patients, but Yoshino et al. reported that 5SCD were also elevated by the intake of mushroom or agaricus (Agaricus subrufescens) [117].

Lai, C.F.; Kao, T.W.; Tsai, T.F.; Chen, H.Y.; Huang, K.C.; Wu, M.S.; Wu, K.D. Quantitative comparison of skin colors in patients with ESRD undergoing different dialysis modalities. Am. J. Kidney Dis. 2006, 48, 292–300. Airaghi, L.; Garofalo, L.; Cutuli, M.G.; Delgado, R.; Carlin, A.; Demitri, M.T.; Badalamenti, S.; Graziani, G.; Lipton, J.M.; Catania, A. Plasma concentrations of α-melanocyte-stimulating hormone are elevated in patients on chronic haemodialysis. Nephrol Dial Transplant. 2000, 15, 1212–1216. Nguyen-Khoa, T.; Massy, Z.A.; De Bandt, J.P.; Kebede, M.; Salama, L.; Lambrey, G.; Witko-Sarsat, V.; Druek, T.B.; Lacour, B.; Thevenin, M: Oxidative stress and haemodialysis: role of inflammation and duration of dialysis treatment. Nephrol Dial Transplant, 2001, 16, 335–340. Köken, T.; Serteser, M.; Kahraman, A.; Gökçe, Ç.; Demir, S. Changes in serum markers of oxidative stress with varying periods of haemodialysis. Nephrology, 2004, 9, 77–82. Shirai, S.; Ominato, M.; Shimazu, T.; Toyama, K.; Ogimoto, G.; Fujino, T.; Yasuda, T.; Sato, T.; Maeba,T.; Owada, S.; Kimura, K. Imbalance between production and scavenging of hydroxyl radicals in patients maintained on hemodialysis. Clin. Exp. Nephrol. 2005, 9, 310–314. Murakami, K.; Wakamatsu, K.; Nakanishi, Y.; Takahashi, H.; Sugiyama, S.; Ito, S. Serum levels of pigmentation markers are elevated in patients undergoing hemodialysis. Blood Purif. 2007, 25, 483-489. Sutherland, W.H.; Gieseg, S.P.; Walker, R.J.; de Jong, S.A.; Firth, C.A.; Scott, N. Serum protein-bound 3,4-dihydroxyphenylalanine and related products of protein oxidation and chronic hemodialysis. Ren Fail, 2003, 25, 997–1009. Murakami, K.; Nakanishi, Y.; Wakamatsu, K.; Yamamoto, K.; Kohriyama, N.; Hasegawa, M.; Tomita, M.; Nabeshima, K.; Hiki, Y.; Asano, S.; Kawashima, S.; Ito, Y.; Fujita, Y.; Asada, H.; Nakai, S.; Sugiyama, S.; Ito, S. Serum levels of 5-S-cysteinyldopa are correlated with skin colors in hemodialysis patients but not in peritoneal dialysis patients. Blood Purif. 2009, 28, 209-215. Bennett, D.C.; Lamoreux, M.L. The color loci of mice: a genetic century. Pigment Cell Res. 2003, 16, 333–344. Kadekaro AL, Wakamatsu K, Ito S, Abdel-Malek ZA: Cutaneous photoprotection and melanoma susceptibility: reaching beyond melanin content to the frontiers of DNA repair. Front Biosci. 2006, 11, 2157–2173. Nagano, A.; Watanabe, A.; Nonaka, Y.; Koga, T.; Wakamatsu, K.; Ito, S.; Nakayama, J. A case of malignant melanoma increasing during pregnancy: measurement of tissue 5-S-cysteinyldopa. Jpn. J. Dermatol. 1999, 109, 1621-1626. Yoshino, K.; Aoki, M.; Kawana, S. Elevation of serum 5-S-CD in a patient with malignant melanoma probably due to Agaricus blazei Murill. Jpn. J. Clin. Dermatol. 2005, 59, 1013-1015.

2. I added newly the following sentences in 8. Discussion and Future perspectives.

As the pathways for the future progress of this 5SCD marker, it is necessary to examine the advanced technology to evaluate the correlation between cut-off values with LDH, S100, MIA, leptin, and so on, and to evaluate as much circulating markers as possible from serum and urine. In addition, the measurement of samples from other than serum or urine, for example, in the cerebrospinal fluid (CSF) or 5SCD levels of BRAF negative versus BRAF positive patients may be a new development. Shinno et al. measured 5SCD levels in CSF in a patient with a variant of neurocutaneous melanosis with leptomeningeal melanoma and reported the clinical significance of 5SCD [118]. Combined chemoimmunotherapy was repeated every other month with monitoring of 5SCD level in CSF. 5SCD levels in CSF have not been monitored in patients with neurocutaneous melanosis, so the basal 5SCD levels in CSF were unclear. By measuring 5SCD levels in CSF collected from patients from Parkinson disease, hemifacial spasm, and moyamoya disease, the basal 5SCD in CSF was determined as 0 nmol/L. 5SCD level decreased after each treatment, but the basal level steadily increased prior to the next treatment. Their findings suggested that periodic measurement of 5SCD levels in CSF is valuable for evaluating the therapeutic efficacy of chemoimmunotherapy in patients with malignant leptomeningeal melanoma and for monitoring the progression of the melanoma. On the other hand, Wakamatsu et al. measured 5SCD level in CSF from one patient. 5SCD levels in serum and CSF were 11.0 nmol/L and 513 nmol/L, respectively (unpublished data). The patient was hospitalized by suspicion of meningoencephalitis in the neurology department. The infection was negative, and melanoma was suspected by a cytology of CSF and consulted by a dermatologist. Therefore, 5SCD determination in serum and CSF was probably thought to be contributed to the diagnosis. There was no primary in skin, and no visceral metastasis, and melanoma was diagnosed as unknown primary region at that time. It was thought to be a melanoma originating in the central nervous system. Circulating cell-free microRNAs (miRNAs) are thought to be involved in the regulation of gene expression and vary with cancer progression or therapeutic response [119]. Using serum samples from 33 patients with malignant melanoma before anti-PD-1 antibody therapy (nivolumab or pembrolizumab), Nakamura et al. measured several miRNAs [120]. Their results suggest that miR-16-5p, miR-17-5p, and miR-20a-5p are useful indicators of successful anti-PD-1 antibody therapy. Furthermore, they proposed that evaluating serum miRNA levels may be a simple method to predict the effectiveness of immune checkpoint inhibitors, although the numbers of patients evaluated are small. To determine whether these three miRNAs can be worthy biomarkers for anti-PD-1 therapy, further studies with a higher number of patients are required.

BRAF/MEK inhibitors are generally considered to be equally effective whether given before or after immunotherapy. In BRAF mutated patients with normal LDH, first-line immunotherapy seems a more effective approach. Czarnecka et al. has demonstrated that the use of two different systemic treatment modalities allows the achievement of comparable survival in BRAF mutated and BRAF wild-type patients although BRAF mutation is a negative prognostic factor in stage IV melanoma [121].

Koh et al. described about Primary malignant melanoma of the breast (PMMB) [122]. PMMB is a rare tumor with only a few case reports available in the literature. They reported two cases of PMMB, one derived from the breast parenchyma and the other from the breast skin. The first case consisted of atypical epithelioid cells without overt melanocytic differentiation like melanin pigments. The tumor cells showed diffuse positivity for S100 protein, tyrosinase, and BRAF V600E. However, the tumor cells were negative for cytokeratin, epithelial membrane antigen, and human melanoma black-45 (HMB-45). The second case showed atypical melanocytic proliferation with heavy melanin pigmentation. The tumor cells were positive for S100 protein, HMB-45, tyrosinase, and BRAF V600E. Although there is little evidence for breast in these two cases, these two cases represent two distinct presentations of PMMB in terms of skin involvement, melanin pigmentation, and HMB-45 positivity.

Shinno, K.; Nagahiro, S.; Uno, M.; Kannuki, S.; Nakaiso, M.; Sano, N.; Horiguchi, H. Neurocutaneous melanosis associated with malignant leptomeningeal melanoma in an adult: clinical significance of 5-S-cysteinyldopa in the cerebrospinal fluid. Neurol. Med. Chir. (Tokyo), 2003, 43, 619-625. Polini, B.; Carpi, S.; Romanini, A.; Breschi, M.; Nieri, P.; Podesta, A. Circulating cell-free microRNAs in cutaneous melanoma staging and recurrence or survival prognosis. Pigment Cell Melanoma Res. 2019, 32, 486–499. Nakahara, S.; Fukushima, S.; Okada, E.; Morinaga, J.; Kubo, Y.; Tokuzumi, A.; Matsumoto, S.; Tsuruta-Kadohisa, M.; Kimura, T.; Kuriyama, H.; Miyashita, A.; Kajihara, I.; Jinnin, M.; Ihn, H. MicroRNAs that predict the effectiveness of anti-PD-1 therapies in patients with advanced melanoma. J. Dermatol. Sci. 2019. doi:10.1016/j.jdermsci.2019.11.010. Czarnecka, A. M.; Teterycz, P.; Mariuk-Jarema, A.; Lugowska, I.; Rogala, P.; Dudzisz-Sledz, M.; Swaitaj, T.; Rutkowski, P. Treatment sequencing and clinical outcomes in BRAF-positive and BRAF-negative unresectable and metastatic melanoma patients treated with new systemic therapies in routine practice. Target. Oncol. 2019, 14, 729-742. Koh, J.; Lee, J.; Jung, S. Y.; Kang, H. S.; Yun, T.; Kwon, Y. Primary malignant melanoma of the breast: a report of two cases. J. Pathol. Transl. Med. 2019, 53, 119-124.